

# Uptake selectivity of Methanesulfonic Acid (MSA) on fine particles over polynya regions of the Ross Sea, Antarctica

Jinpei Yan[*1,2], Jinyoung Jung[3], Miming Zhang[1,2], Federico Bianchi[4], Yee Jun Tham[4], Suqing Xu[1,2], Qi Lin[1,2], Shuhui Zhao[1,2], Lei Li[5], Liqi Chen[1,2]

*1 Key Laboratory of Global Change and Marine-Atmospheric Chemistry, Xiamen 361005, China;*

*2 Third Institute of Oceanography, Ministry of Natural Resources, Xiamen 361005, China;*

*3 Korea Polar Research Institute, 26 Songdomirae-ro, Yeonsu-gu, Incheon 21990, Republic of Korea;*

*4 Institute for Atmospheric and Earth System Research; University of Helsinki, 00014, Finland;*

*5 Institute of Mass Spectrometer and Atmospheric Environment, Jinan University, Guangzhou 510632, China*

***Abstract:*** The uptake of methanesulfonic acid (MSA) on existing particles is a major route of the particulate MSA formation, however, MSA uptake on different particles is still lack of knowledge. Characteristics of MSA uptake on different aerosol particles were investigated in polynya regions of the Ross Sea, Antarctica. Particulate MSA mass concentrations, as well as aerosol populations and size distributions, were observed simultaneously for the first time to access the uptake of MSA on different particles. The results showed that MSA mass concentration did not always reflect MSA particle population in the marine atmosphere. MSA uptake on aerosol particles increased the particle size and changed aerosol chemical compositions, but did not increase the particle population. The uptake rates of MSA on existing particles were significantly influenced by aerosol chemical properties. The favor uptake of MSA occurred on the sea salt particles, as MSA-Na and MSA-Mg particles were abundant in the Na and Mg particles, accounting for $0.43 \pm 0.21$ and $0.41 \pm 0.20$ of the total Na and Mg particles, respectively. However, acidic and hydrophobic particles suppressed the MSA uptake, as MSA-EC and MSA-$SO_4^{2-}$ accounted only $0.24 \pm 0.68$ and $0.26 \pm 0.47$ of the total EC and $SO_4^{2-}$ particles, respectively. The results extended the knowledge of the formation and environmental behavior of MSA in the marine atmosphere.

***Keywords****:* Methanesulfonic acid (MSA); nss-$SO_4^{2-}$; aerosol; climate change; Antarctica

## 1. Introduction

---

*Corresponding author. Tel.: +86 592 2195370; Fax: +86 592 2195280, E-mail address: jpyan@tio.org.cn; Address: No.178 Daxue Road, Siming district Xiamen, Third Institute of Oceanography, MNR, 361005, P R China.



25  Methanesulfonic acid (MSA) and non-sea-salt-sulfate (nss-$SO_4^{2-}$), deriving from the oxidation

26  of dimethyl sulfide (DMS), are important sources of cloud condensation nuclei (CCN) in the

27  marine boundary layer (Chang et al., 2011; Ghahremaninezhad et al., 2016). Different from

28  nss-$SO_4^{2-}$, MSA is exclusively from the oxidation of DMS in the atmosphere (Sorooshian et al.,

29  2007). Thus, MSA was expected as a useful marker for the deconvolution of sulfate from marine

30  biogenic and non biogenic sources (Legrand et al., 1998).The ratio of MSA to nss-$SO_4^{2-}$ is often

31  used to assess the DMS oxidation routes and the contributions of biogenic sulfur to other sulfur

32  sources (Sorooshian et al., 2007; Wang et al., 2014). DMS oxidation routes, as well as the

33  products of MSA and nss-$SO_4^{2-}$, have been investigated previously in the marine atmosphere

34  (Preunkert et al., 2008; Kloster et al., 2006).

35  Generally, particulate MSA was generated from the reactive uptake of DMS and condensation of

36  gaseous MSA on aerosol particles (Davis et al., 1998; Barnes et al., 2006). A recent study showed

37  that MSA may increase sulfate cluster formation rate by up to one order of magnitude, increasing

38  the stability of the clusters (Bork et al., 2014). However, previous studies have showed that $SO_4^{2-}$

39  was more effective at new particle formation (NPF) than MSA, while MSA was more likely to

40  condense onto existing particles (Hayashida et al., 2017). Although the reactive uptake of MSA

41  on fine particles was demonstrated in the previous studies (Sorooshian et al., 2007; Bates et al.,

42  1992), the influence of aerosol characteristics on MSA uptake has not been presented.

43  The chemical components and sources of aerosol particles in the marine atmosphere were

44  rather complicated (Weller et al., 2018). Filtered sample methods were often used in previous

45  studies (Jung et al., 2014; Preunkert et al., 2007; Read et al., 2008) with a long sampling interval

46  to accommodate the detection limit of the instrument (Preunkert et al., 2007; Zhang et al., 2015).

47  It is, therefore, difficult to clarify how MSA mixes with other aerosol species, using bulk aerosol

48  sampling methods, as only the mean aerosol chemical components were presented during the

49  sampling period (Bates et al., 1992; Chen et al., 2012). On-line aerosol mass spectrometry has

50  been used to characterize the aerosol chemical species and sizes with high-time-resolution (Yan et

51  al., 2018; Healy et al., 2010), allowing the determination of particle mixing states and sources.

52  Although a few studies have shown that MSA was often associated with Mg in aerosol particles,

53  probably due to marine biogenic activity (Casillas-Ituarte et al., 2010), studies of the interactions





between MSA and other aerosol species were rare. Theoretical and laboratory studies have attempted to explain these observations and determine in which state MSA enters the aerosol particles (Bork et al., 2014). However,the relative likelihood of MSA uptake on different particles remains uncertain.

In this study, we examined the uptake characteristics of MSA on different particles over polynya regions in the Ross Sea (RS), Antarctica, based on high-time-resolution observations. MSA mass concentrations and particle populations, as well as aerosol compositions and size distributions, were measured simultaneously for the first time in the RS using an in-situ gas and aerosol compositions (IGAC) and a single particle aerosol mass spectrometer (SPAMS) monitoring instruments. Observations were carried out in two different seasons, the early December with intensity sea ice coverage and in the mid-January to February with sea ice free in the RS. The selectivity of MSA uptake on different particles was investigated in the RS with different circumstances.

## 2. Experiment methods and observation regions

The observations were carried out on-board of R/V "Xuelong", covering a large region of the RS, Antarctica (50°S to 78°S, 160°E to 185°E) (Fig. S1) with different sea ice concentrations. The leg I was carried out from December 2 to 20, 2017. The sea surfaces were covered with intense sea ices in the RS during this period (Fig. S4a). However, when we arrived back in the RS (leg II, from January 13 to February 14, 2018), the sea ices have almost melted in the RS (Fig. S4b).

### 2.1 Observation instruments and sampling inlet

An in-situ gas and aerosol compositions monitoring system (IGAC, Model S-611, Machine Shop, Fortelice International Co., Ltd., Taiwan; http://www.machine-shop.com.tw/), and a single particle aerosol mass spectrometer (SPAMS, Hexin Analysis Instrument Co., Ltd.) were used to determine aerosol water-soluble ion species, particle size distributions and chemical compositions,



respectively (Fig. S2). The sampling inlet connecting to the monitoring instruments was fixed to a
mast 20 meters above the sea surface. A total suspended particulate (TSP) sampling inlet was
positioned at the top of the mast. Conductive silicon tubing with an inner diameter of 1.0 cm was
used to make the connection to all instruments.
**2.2 Aerosol water-soluble ion species**
Gases and aerosol water-soluble ion species were determined using a semi-continuous IGAC
monitor. Gases and aerosols were separated and streamed into a liquid effluent for on-line
chemical analysis at an hourly temporal resolution (Young et al., 2016; Liu et al., 2017). The
analytical design and methodology for the determination of gases and aerosol water-soluble ions
have been described in detail by Tao (2018) and Tian (2017). Fine particles were firstly enlarged
by vapor condensation and subsequently accelerated through a conical-shaped impaction nozzle
and collected on the impaction plate. The samples were then subsequently analyzed for anions and
cations by an on-line ion chromatography (IC) system (DionexICS-3000). The injection loop size
was 500 μL for both anions and cations. Six to eight concentrations of standard solutions were
selected for calibration, depending on the target concentration, in which the $R^2$ was above 0.997
(Fig. S3). The detection limits for $MSA^-$, $SO_4^{2-}$, $Na^+$, and $Cl^-$ were 0.09, 0.12, 0.03, and 0.03 μg/L
(aqueous solution), respectively.
**2.3 Aerosol size distribution and chemical compositions**
The detection method for fine particles using a SPAMS has been described in detail by Li (Li et
al., 2011; Li et al., 2014). Particles were introduced into the vacuum system through a critical
orifice, then focused and accelerated to form a particle beam with specific velocity. The particle
beam was passed through two continuous diode Nd: YAG lasers (532 nm), where the scattered
light was detected by two Photomultiplier Tubes (PMTs). The velocity of a single particle was





then determined and converted into its aerodynamic diameter. The individual particle was ionized
with a 266 nm Nd: YAG laser to produce positive and negative ions. The fragment ions were
analyzed using a bipolar time-of-flight mass spectrometer. The power density of the ionization
laser was kept at $1.56 \times 10^8$ w/cm$^2$.

The particle size data and mass spectra were analyzed using the YAADA software toolkit

(http://www.yaada.org/) (Allen 2005). An adaptive resonance theory based neural network
algorithm (ART-2a) was applied to cluster individual particles into separate groups based on the
presence and intensity of ion peaks in the single particle mass spectrum (Song et al., 1999), with a
vigilance factor of 0.65, a learning rate of 0.05, and a maximum of 20 iterations.
**2.4 Metrological data**

Meteorological parameters such as temperature, humidity, wind speed, and direction were

measured continuously using an automated meteorological station deployed in the R/V "Xuelong",
which was located on the top deck of the vessel.
**2.5 Satellite data of sea ice and chlorophyll-a**

In this study, we used remote sensing data to show the spatial and temporal distribution of

chlorophyll and sea ice concentrations in the study region. Due to the cloud effect and swath limits,
we chose the 8-day datasets for the remote sensing of chlorophyll-a from MODIS-Aqua
(http://oceancolor.gsfc.nasa.gov) with a spatial resolution of 4 km. We used the sea ice on
centration data from the daily 3.125-km AMSR2 dataset (Spreen et al., 2008) (available at
https://seaice.uni-bremen.de). Each grid of the gridded datasets with a sea ice concentration less
than or equal to 15 % was regarded as comprising all water (Cavalieri et al., 2003). The time series
of the total regional mean value in the study region was then plotted.
**3. Results and discussion**
**3.1. Spatial distributions of MSA mass concentration and particle population**

MSA mass concentrations and their populations were measured continuously in the RS. MSA

concentrations ranged from 14.6 to 210.8 ng.m$^{-3}$, with an average of $43.8 \pm 22.1$ ng.m$^{-3}$ during leg



I (Fig.1a), consisting with summertime MSA levels recorded at Halley station, averaging 35.3
ng.m$^{-3}$ (75$^o$39'S) and Dumont d'Urville station (66$^o$40'S) (Minikin et al., 1998), averaging 49
ng.m$^{-3}$, but lower than those reported at Palmer station, averaging 122 ng.m$^{-3}$ (64$^o$77' S) (Savoie et
al., 1993). The highest MSA levels occurred at the region (64$^o$ - 67$^o$ S), with an maximum value of
210.8 ng.m$^{-3}$ (Fig. 1a), consisting with the previous observation results from the Southern Ocean
(60 - 70$^o$ S; maximum MSA level of 260 ng.m$^{-3}$) (Chen et al., 2012). In this study, elevated MSA
levels were associated with the dynamic sea ice edge at ~64$^o$ S, once the sea ice started to melt in
the early December (Fig. S4a and Fig. S4c). The release of iron (De Baar et al., 1995; Wang et al.,
2014) and algae (Lizotte et al., 2001; Loose et al., 2011) from sea ice increased phytoplankton
numbers (Taylor et al., 2013), resulting in the increase of DMS generation and emission
(Hayashida et al., 2017). This, in turn, increased MSA levels due to the oxidation of DMS in the
atmosphere.
MSA particle populations (0.1 - 2.5 μm) were determined simultaneously by SPAMS during the
leg I (Fig. 1b). The highest average hourly MSA particle population (507 $\pm$ 189) occurred at
MP1 (68$^o$ - 72$^o$S, 172$^o$E) near the Antarctic continent, following by MP2 (65$^o$ - 68$^o$S, 160$^o$ - 170$^o$
E), with an average particle population of 344 $\pm$ 334. High MSA particle populations were
associated with high wind speeds in these regions (MP1 8.06 ± 1.86m/s; MP2 15.76 ± 3.93m/s;
Fig. 2).
The MSA concentration ranged from 11.4 - 165.4 ng•m$^{-3}$ (with an average of 38.8 ± 27.5 ng•m$^{-3}$)
during leg II, and the MSA particle population ranged from 3 - 1666 (with an average of 168 ±
172; Fig. 1c and 1d). Extremely high MSA levels, with an average of 100.3 ± 18.6 ng•m$^{-3}$, were
observed in the MA region (170.2° - 177.4°E, 68.2° - 77.8°S), but we did not observed high MSA
particle populations in this region (with an average of 171 ± 159). High MSA particle numbers
with low MSA concentrations occurred at MP1 and MP2 (Fig.1a and 1b). It indicated that MSA
mass concentrations did not always reflect the MSA particle populations in the marine atmosphere.
Generally, the uptake of MSA on aerosol surfaces (Read et al., 2008) only changed the aerosol
size and chemical compositions, without varying their population. Hence, the MSA particle
population was mainly associated with the aerosol number in the atmosphere, as more particles
were provided for the uptake of MSA in high particle population. Though high levels of MSA
would also increase the MSA population, high MSA mass concentrations with low MSA
populations were observed in this study. This phenomenon occurred when low existing particle
populations and high MSA mass concentrations were presented in the marine atmosphere.

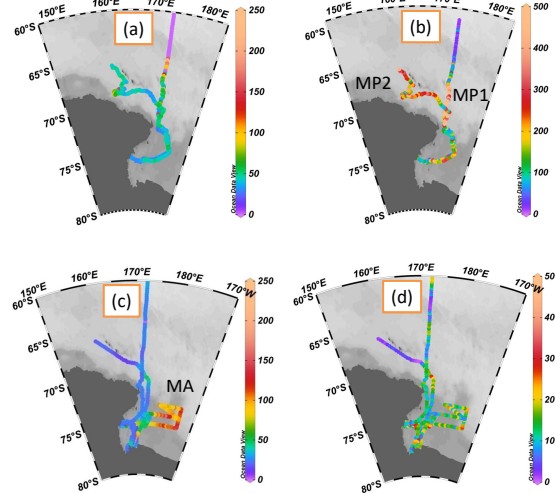


Fig.1 Spatial distribution of MSA mass concentrations and particle populations, (a) MSA mass

concentrations during leg I (ng.m$^{-3}$); (b) MSA particle populations during leg I; (c) MSA mass

concentrations during leg II (ng.m$^{-3}$) and (d) MSA particle populations during leg II.

**3.2. Linkage between MSA concentration and particle population**

To verify the relationship between MSA mass concentration and particle population, the

temporal distributions of MSA mass concentration and particle number are illustrated in Fig. 2.
Variations of MSA mass concentration were not always associated with the MSA particle



population during the observation periods (Fig. 2a and Fig. 2b). MSA particle number did not
show an obvious correlation with MSA mass concentration (Fig. S5a), indicating that the major
factors regulating MSA mass concentration and MSA particle population were different. High
MSA particle populations often occurred in conjunction with high wind speeds (Fig. 2b and 2d),
while high MSA mass concentrations were not always observed at high wind speed regions, such
as extremely high MSA mass concentrations with low wind speeds were presented at MA (Fig. 2a,
2b and 2d).
MSA mass concentrations were determined by the oxidation of DMS, derived from marine
phytoplankton activity (Davis et al., 1998; Barnes et al., 2006; Read et al., 2008), but MSA
particle populations were mainly associated with the uptake of MSA on existing particles. High
existing particle populations led to high MSA particle populations, as the formation of particulate
MSA often occurred on the surfaces of existing particles (Read et al., 2008). In this study, the
variation of MSA particle population consisted with the variation of total particle population
during the observation period (Fig. 2b). A strong positive correlation between MSA particle
population and total particle population were presented (slope=0.19, $r^2$=0.65, n=1195, Fig. S5b).
The ratio of MSA particle population to total particle population ($R_{MSA/total}$) concentrated on the
range of 0.2 - 0.5, with an average of $0.29 \pm 0.15$ (Fig. 2b).

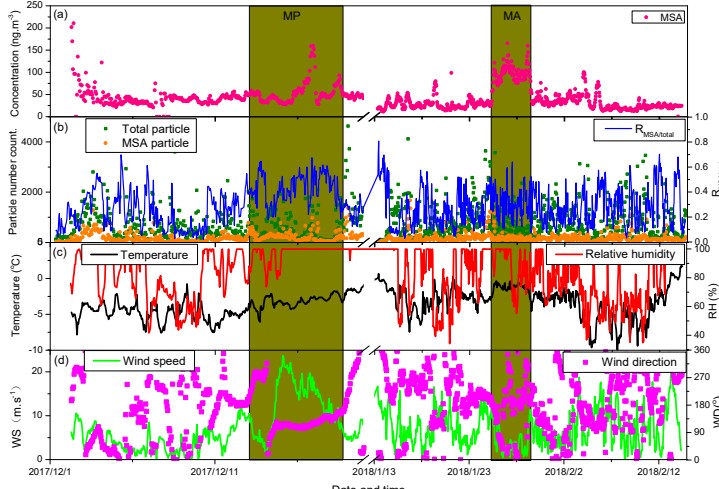


Fig. 2 Relationship between MSA mass concentration and MSA particle population in the context
of various environmental factors. (a) Time series of MSA mass concentrations; (b) Time series of





MSA particle population, total particle population and the ratio of MSA particle population and
total particle population; (c) Time series of temperature and relative humidity and (d) Time series

of wind speeds and direction.

**3.3. Signatures of MSA particle types**

During leg I, 332438 single particles with positive and negative mass spectra were obtained,

while 603098 single particles with positive and negative mass spectra were obtained during leg II.
MSA particles, identified during the cruise using the ART-2a algorithm (Song et al., 1999),
accounted for 27.69 % and 22.08 % of the total particles during leg I and leg II, respectively. To
investigate the interactions between MSA and other species, MSA particles were further classified
into seven sub-types, including MSA-Na, MSA-Mg, MSA-$SO_4^{2-}$, MSA-K, MSA-EC, MSA-OC,
and MSA-$NO_x^-$.
**3.3.1 MSA-Na particles**

Sodium, which is often associated with sea salt particles in the marine atmosphere (Teinila et

al., 2014), is an important component of atmospheric aerosols in ocean regions (Yan et al., 2018).
Fig. 3a illustrates the average mass spectra of MSA-Na particles during leg I and leg II. Strong
$Na^+$ peaks with weak $K^+$, $Ca^+$, and $Na_2Cl^+$ peaks were observed in the positive spectrum, while
strong $NaCl_2^-$ and $MSA^-$ peaks with low $Cl^-$, $HSO_4^-$, $NO_3^-$, and $O^-$ peaks were observed in the
negative spectrum. Similar average mass spectra for MSA-Na particles were observed during leg I
and leg II, even though the two measurements were carried out under different circumstances.
MSA-Na particles were the most dominant type of MSA particles, accounting for more than 30 %
of total MSA particles (Fig. 4).
**3.3.2 MSA-Mg particles**

209        Mg is another common component in ocean-derived particles, hence, such particles are often

classified as sea salt particles in the marine atmosphere. However, some previous studies have





shown that the chemical properties of Mg particles observed in marine environment were distinct
from those of sea salt particles (Gaston et al., 2011). In this study, the mass spectral characteristics
of MSA-Mg type particles included strong $MSA^-$ and Mg peaks (Fig. 3b). In sea salt particles, the
dominant peak was typically $Na^+$ rather than $Mg^+$ (Fig. 3a) due to the higher concentration of $Na^+$
in seawater (Guazzotti et al., 2001). Similar with MSA-Na type particles, strong $Na^+$ and $NaCl_2^-$
peaks with weak $Cl^-$, $NO_3^-$, $K^+$, and $Ca^+$ peaks were observed in the mass spectra, indicating that
MSA-Mg type particles were also derived from sea salt particles. Strong positive correlation
($r^2$=0.95) between MSA-Na and MSA-Mg was presented in this study (Fig. S6), indicating that
these two types of particles were derived from the same sources. However, the abundance of $Mg^+$
fragment ion relative to $Na^+$ fragment ion in MSA-Mg type particles was different from MSA-Na
type particles, indicating that MSA-Mg particles were also affected by other sources. Studies have
shown that Mg particles were correlated strongly with atmospheric DMS ($r^2$=0.76) (Gaston et al.,
2011), indicating that Mg particles were also impacted by marine biological materials, such as cell
debris or fragments, viruses, bacteria, or the organics released by lysed cells (Casillas-Ituarte et al.,
2010; Gaston et al., 2011). Hence, MSA-Mg type particles were associated with both sea salt
particles and biological emissions.
**3.3.3 MSA-$SO_4^{2-}$ particles**
$SO_4^{2-}$ may be derived from different sources, such as sea salt aerosols, anthropogenic
emissions, marine biogenic and volcanic sources (Legrand et al., 1998). Strong signals, peeking at
m/z -97 $HSO_4^-$ and m/z -95 $MSA^-$, were presented in the negative spectrum (seen in Fig. 3c),
consisting with previous studies with intense signals of $HSO_4^-$ and $MSA^-$ occurred at m/z -97 and
m/z -95 (Gaston et al., 2011; Silva et al., 2000). The simultaneous present of $K^+$, $Na^+$, $Al^+$, and $Fe^+$





peaks in the positive mass spectrum and $NaCl_2^-$, $NO_3^-$, $C_4H^-$ and $C_2H_2^-$ peaks in the negative mass
spectrum, suggesting that MSA-$SO_4^{2-}$ particles were associated with different sources. This can be
further demonstrated by the size distribution of MSA-$SO_4^{2-}$ particles (Fig. 5), as MSA-$SO_4^{2-}$
particles are found in both fine and coarse particles.
**3.3.4 MSA-K particles**
The positive mass spectrum of the MSA-K particles was dominated by a strong $K^+$ peak with
weak $Na^+$, $C_2H_3^+$ and $C_3H_7^+$ peaks (Fig. 3d). Strong $HSO_4^-$, and $MSA^-$ signals were presented in
the negative mass spectrum. Abundance of organic ion fragments were observed in the mass
spectra of MSA-K particles. Generally, K was expected as a marker of biomass-burning source in
continental areas (Yan et al., 2018). However, the mass spectra of MSA-K particles observed here
were very different from the mass spectra of K particles observed in continental areas, suggesting
that K particles from marine sources were quite different from continental sources.
**3.3.5 MSA-OC particles**
OC particles are often associated with anthropogenic sources, such as vehicle and coal
combustion (Silva et al., 2000; Stiaras et al., 2008), marine biogenic sources (Quinn et al., 2014)
and secondary sources (e.g. photochemical reaction from their precursor organic gases) (Horne et
al., 2018). The positive and negative mass spectra of MSA-OC were dominated by $C_xH_y$ ion peaks
(i.e., $C_2H_3^+$, $C_3H^+$, $C_3H_3^+$, $C_3H_4^+$, and $C_3H_7^+$; Fig. 3e). Strong signals of $HSO_4^-$ and $MSA^-$ fragment
ions were also presented in the negative spectrum, while a few signals of $Na^+$ and $Cl^-$ were
observed in the positive mass spectrum (Fig. 3e).
**3.3.6 MSA-EC particles**
EC particles are often associated with primary emissions; that is, the incomplete combustion





of carbon-containing materials (Murphy et al., 2009). In this study, MSA-EC particles were
characterized by strong peaks of $C_n^-$ ($C_4^-$, $C_3^-$ and $C_2^-$) in the negative spectrum, while the positive
mass spectrum were dominated by $Ca^+$ ions (Fig. 3f). Compare with the average mass spectra of
MSA-OC particles, the abundances of $MSA^-$ and $HSO_4^-$ fragment ions were lower in MSA-EC
particles, indicating that the uptake of MSA on EC particles might be more difficult than the
uptake of MSA on OC particles. Similar with the mass spectra of MSA-OC particles, a few
fragments of $Na^+$ and $Cl^-$ were observed in the MSA-EC mass spectra, suggesting that MSA-OC
and MSA-EC particles rarely mixed with sea salt particles.
**3.3.7 MSA-$NO_x^-$ particles**
The negative spectrum of MSA-$NO_x^-$ particle was dominated by strong peaks of $MSA^-$, $NO_2^-$,
and $NO_3^-$, with weak $C_xH_y^-$, $O^-$, and $Cl^-$ peaks (Fig. 3g). Strong $Na^+$, $(C_3H_3^+)/K^+$, and $(C_3H_4^+)/Ca^+$
peaks with weak $Na_2Cl^+$ and $CaO^+$ peaks were observed in the positive spectrum. Sea salt particles
reacted with atmospheric $HNO_3$ easily to form nitrate and hydrogen chloride (Adachi et al., 2015).
The abundance of $Na^+$, $Cl^-$, and $NaCl_2^-$ ions in the mass spectra of the MSA-$NO_x^-$ particles
demonstrated these particles were formed by the interaction between sea salt particles and $NO_x^-$ in
the marine atmosphere.

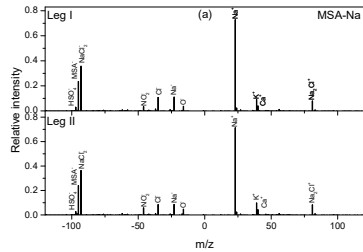
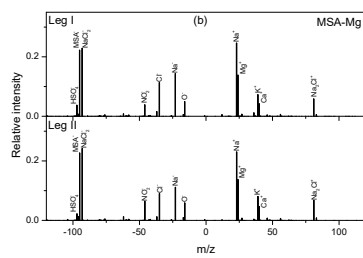




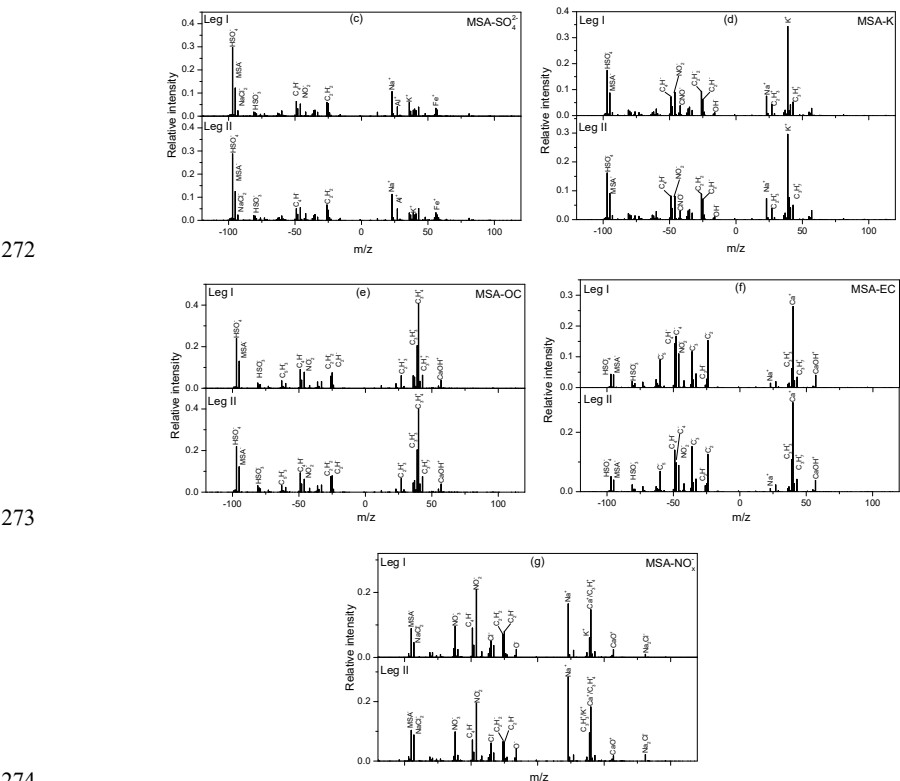

Fig. 3 Average mass spectra of major MSA clusters during leg I and leg II. (a) MAS-Na, (b) MSA-Mg, (c) MSA-$SO_4^{2-}$, (d) MSA-K, (e) MSA-OC, (f) MSA-EC, and (g) MSA-$NO_x^-$.

**3.4 Uptake characteristics of MSA on existing particles**

In this study, Na, Mg, and $SO_4^{2-}$ were the most abundant particles (Fig. 4a). Similar with Na, Mg, and $SO_4^{2-}$, MSA-Na, MSA-Mg, and MSA-$SO_4^{2-}$ were also the three most abundant MSA particles (Fig. 4b), accounting for more than 70 % of the total MSA particles. It indicated that the uptake of MSA was associated with the particle population. However, we found that $SO_4^{2-}$ was the most abundant particles of the total aerosol particles, while MSA-$SO_4^{2-}$ was not the most abundant MSA particles in the atmosphere. The results revealed that the formation of particulate MSA on existing particles was affected by other factors, except particle population.

The average fractions of the MSA particle sub-types differed considerably from the average

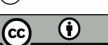



fractions of their corresponding particle types (Fig. 4c and 4d). $SO_4^{2-}$ was the most abundant
particles, accounting for 26.8% of the total particles (Fig. 4c). However, MSA-$SO_4^{2-}$ particles
accounted for only about 17.8% of the total MSA particles (Fig. 4d). Similarly, the relative
abundances of MSA-EC and MSA-K with respect to total MSA particles were lower than those of
EC and K with respect to the total particles. In contrast, MSA-Na particles were the most abundant
MSA particles, accounting for more than 32.55% of the total MSA particles (Fig. 4d), while Na
particles accounted for only 21.68% of the total particles (Fig. 4c). Similar patterns were observed
for Mg and OC particles. MSA-Mg and MSA-OC particles were more abundant in the MSA
particles than Mg and OC particles in the total particles (Fig. 4d). These results indicated that the
uptake of MSA on Na and Mg particles was more favorable than the uptake of MSA on EC and
$SO_4^{2-}$ particles. Note that the observations during leg I and leg II were performed under different
circumstances, as high concentrations of sea ice were presented during leg I (Fig. S4a) but sea ice
free was presented during leg II (Fig. S4b). Despite different conditions were presented during leg
I and leg II, the relative fractions of MSA-type particles remained similar, conforming the uptake
selectivity of MSA occurred on different particles.

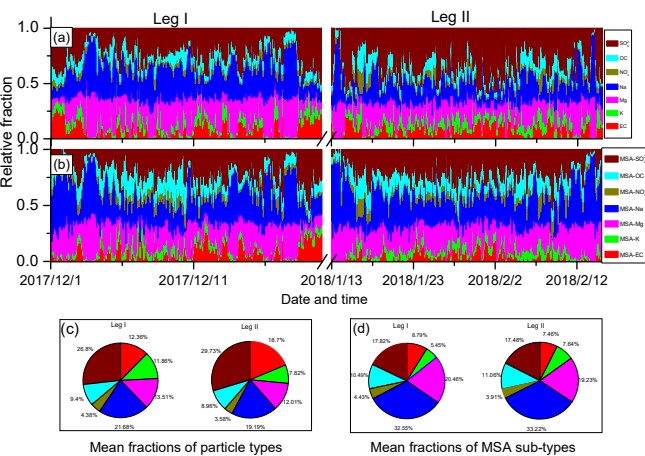



Fig. 4 Relative fractions of different particle types during leg I and leg II. (a) Relative fractions of
different particle types; (b) Relative fractions of MSA-type particles; (c) Average fractions of
different particle types during leg I and leg II; and (d) Average fractions of MSA-type particles
during leg I and leg II.

The size distributions of total particles and MSA particles are illustrated in Fig. 5. The size of
the total particles showed an unimodal distribution, with a mean diameter of 0.51 μm, during leg I
and leg II (Fig. 5a). Most of the particles were 0.3-1.0 μm, consisting with particle sizes observed in
Antarctica using SMPS (Pant et al., 2011), and with sea spray aerosol sizes measured in marine
regions (Quinn et al., 2017). Compare with mean size of the total particles, MSA particles were larger
in this study, with a mean diameter of 0.65 μm (Fig. 5a). This suggested that particles were enlarged
when MSA uptake occurred on their surfaces. Although particles were enlarged by MSA uptake,
submicron MSA particles represented more than 90 % of the total MSA particles (Fig. 5b), indicating
that most of the MSA particles were still in the submicron range, consisting with observation results in
coastal Antarctica (Legrand et al., 1998) and the Pacific Ocean (Jung et al., 2014).
The size-resolved MSA sub-type particles by population fraction during leg I and leg II were
also given in this study. MSA-EC, MSA-K and MSA-$NO_x^-$ particles were primarily distributed in
small size (<1 μm) (Fig. 5c). In contrast, high percentage of MSA-Na and MSA-Mg particles were
presented in large particles (>1 μm), accounting for more than 75 % of the total coarse particles
(Fig. 5c). However, MSA-$SO_4^{2-}$ and MSA-OC particles had a wide size distribution, mainly due to
the variety sources for these two types of particles. In this study, $SO_4^{2-}$ was mainly derived from
sea salt particles and the oxidation of DMS. Sea salt aerosols generated by whitecaps and bursting
bubbles, which were generally in the coarse mode (Norris et al., 2013), while $SO_4^{2-}$ particles
generated from the oxidation of DMS were mainly distributed in submicron ranges (Legrand et al.,

1998).

326  Although the MSA particle and total particle populations during leg II were much higher than

327 during leg I (Fig. 5a) and seasonal conditions were different between leg I and leg II (Fig. S4), the

328 size-resolved MSA sub-type particles identified during leg II were very similar with the

329 size-resolved MSA sub-type particles identified during leg I (Fig. 5c and 5d), confirming the

330 stable MSA uptake properties on different particles.

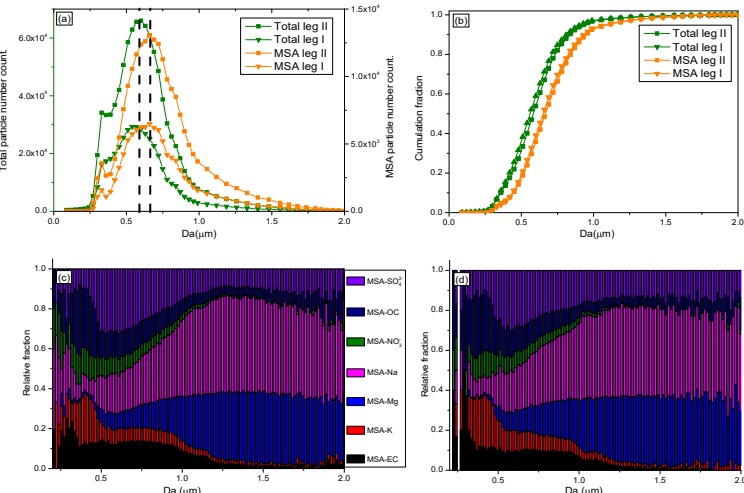

332   Fig. 5 Size distributions of MSA particles and size-resolved MSA sub-type particles during the

333 cruise, (a) Size distributions of MSA particles and total particles, (b) Cumulative size distributions

334    of MSA and total particles, (c) Size-resolved MSA sub-type particles during leg I, and (d)

335        Size-resolved MSA sub-type particles during leg II.

336 **3.5. The uptake rate of MSA on different particles**

337   The uptake of MSA on the existing particles had been investigated, however, the pressing

338 question was how different aerosol properties impacted the uptake rates of MSA. Fig. 6 shows the

339 uptake rates of MSA (defined as the ratio of MSA-containing particles to the corresponding

340 particles, such as MSA-Na to Na ratio) on different particles in marine atmosphere. The formation

341 of particulate MSA included two routes, the reactive uptake of DMS on existing aerosols, and the

342 conversion of gaseous MSA to particulate MSA through condensation on existing particles (Read

343 et al., 2008). High uptake rates of MSA-Na and MSA-Mg particles were observed in Na and Mg

344 particles, accounting for $0.43 \pm 0.21$ and $0.41 \pm 0.20$ of the total Na and Mg particles,

345 respectively (Fig. 6). There were two reasons for the effective uptake of MSA on Na and Mg





particles. Generally, Na and Mg particles are mainly derived from sea salt particles, which are
often alkaline. Previous studies have shown that alkaline sea salt particles were favor to absorb
acidic atmospheric gases, promoting the formation of acidic compounds on sea salt particles
(Laskin et al., 2003). As an acidic species, MSA was easily absorbed by sea salt particles to form
particulate MSA. The other one halogen radicals on the surfaces of sea salt particles also improved
the oxidative reactive uptake of DMS on sea salt particles to form particulate MSA (Read et al.,

2008).

Low uptake rate ($0.24 \pm 0.68$) of MSA-EC particles was observed in this study (Fig. 6). EC
particles, emitting from fossil fuel combustion, are highly hydrophobic. In this case, it would
suppress the uptake of MSA on EC particles, as DMS reactive uptake often occurred through
aqueous reactions (Bardouki et al., 2003). The relative fraction of $SO_4^{2-}$ particles was much higher
than that of EC particles (Fig. 4c). However, the uptake rate of MSA-$SO_4^{2-}$ particles ($0.26 \pm$
$0.47$) was similar with that of MSA-EC particles (Fig. 6), indicating that particle population was
not the major factor affecting MSA uptake rate. The uptake rate of MSA on existing particles was
significant dependent on particle characteristics. As $SO_4^{2-}$ particles were often acidic, MSA uptake
by this type of particle was restricted. For this reason, even though the $SO_4^{2-}$ particle population
was much higher than Na particle population, the uptake rate of MSA-$SO_4^{2-}$ particles was much
lower than that of MSA-Na particles (Fig. 6).
Following by the MSA-Na and MSA-Mg particles, the uptake rates of MSA-OC and
MSA-$NO_x^-$ particles were $0.37 \pm 0.38$ and $0.35 \pm 0.62$, respectively (Fig. 6). This consisted with
the relative abundances of MSA-OC and MSA-$NO_x^-$ particles and the corresponding OC and $NO_x^-$
particles (Fig. 4). It indicates that the uptake rates of MSA on existing particles were determined
by the aerosol properties, alkaline sea salt particles enhanced the uptake of MSA, while acidic and
hydrophobicity species suppressed the uptake of MSA on these particles. The differences of
uptake rates of MSA on different types of particles extended the knowledge of DMS oxidation and
the formation of particulate MSA in the marine atmosphere.





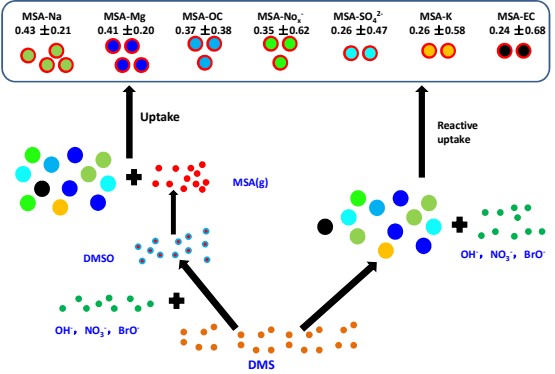


Fig. 6 MSA uptake rates on different aerosol particles in the marine atmosphere.
**4. Conclusions**
The uptake characteristics of MSA on different aerosols were examined during the early
December 2017 and January-February 2018 in the polynya regions of the RS, Antarctica.
Particulate MSA mass concentration, as well as particle populations and size distributions, were
determined simultaneously for the first time to characterize the formation of MSA on different
particles. To access the interactions between MSA and other species, MSA particles were
classified into seven sub-types using the ART-2a algorithm: MSA-Na, MSA-Mg, MSA-$SO_4^{2-}$,
MSA-K, MSA-EC, MSA-OC, and MSA-$NO_x^-$.
MSA mass concentration did not always reflect MSA particle population in the marine
atmosphere. MSA uptake occurred on aerosol surfaces altered the aerosol size and chemical
compositions, but did not change the aerosol population. MSA particle number count was mainly
associated with the total particle number count, as more particles implied a greater opportunity for
MSA uptake. High MSA mass concentrations with low MSA population occurred, when low
existing particle population with high MSA production from the oxidation of DMS were
presented.
The uptake of MSA on existing particles was mainly dependent on aerosol properties. Alkaline
sea salt particles enhanced the uptake of MSA, as high uptake rates of MSA-Na and MSA-Mg
particles were observed in the Na and Mg particles, accounting for 0.43 ± 0.21 and 0.41 ± 0.20 of





the total Na and Mg particles, respectively. But acidic and hydrophobicity species suppressed the
uptake of MSA on these particles, as only $0.24 \pm 0.68$ and $0.26 \pm 0.47$ of MSA-EC and
MSA-SO$_4^{2-}$ were presented in the total EC and SO$_4^{2-}$ particles. The results extend the knowledge
of the impact of aerosol properties on the conversion of MSA in the marine atmosphere, however,
the details of the formation of MSA are complicated and still controversial. Observations and
especially simulation experiments in the laboratory are need in the future to clarify the formation
of MSA and their impact factors in the marine atmosphere.

**Author contributions**

JY conducted the observations, analyzed the results, and wrote the paper. JJ contributed the
data analyses and paper writing. MZ conducted the on-board observations. FB and YT contributed
to the refining the ideas and contributed considerably to the interpretation of the results. SX and
SZ applied the calculations of sea ice distribution and Metrological data. QL and LL contributed
the observation data analyses. LC and JY were together responsible for the design of the study. All
authors were involved in discussing the results and improved the paper by proofreading.

**Acknowledgements**

This study is Financially Supported by Qingdao National Laboratory for marine science and
technology (No. QNLM2016ORP0109), the Natural Science Foundation of Fujian Province,
China (No. 2019J01120), the National Natural Science Foundation of China (No. 21106018, No.
41305133). Jinyoung Jung was supported by grants from Korea Polar Research Institute (KOPRI)
(PE19060). The authors gratefully acknowledge Guangzhou Hexin Analytical Instrument
Company Limited for the SPAMS data analysis and on-board observation technical assistance,
and Zhangjia Instrument Company Limited (Taiwan) for the IGAC technical assistance and data
analysis.

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
