# Peer review of "Uptake selectivity of Methanesulfonic Acid (MSA) on fine particles over polynya regions of the Ross Sea, Antarctica"

_Atmospheric Chemistry and Physics, 2019_

## Referee Comment (RC1)

**Summary/recommendations:**

This is an interesting paper that provides a generally clear and thorough analysis of particulate-phase MSA in the Ross Sea region. The authors have done an excellent job presenting several different results that provide a clear picture of MSA-containing particles in this region for particles between ~0.1-2 μm. I particularly appreciated their discussions upon why MSA condenses on some particle types but not others. This study can become a useful resource to the community. However, there were details missing in the methods that made a complete evaluation of this study difficult. My primary concern is that the authors did not make it clear whether the datasets were screened for ship exhaust (pollution) contamination. If this was not done, the results would likely be skewed, in particular those that discuss particle speciation. I have outlined a few other concerns in my review below. If the authors have corrected their data for ship contamination and simply didn't include these details in the manuscript, then I recommend that this study be published after the revisions discussed below. If the authors have not corrected their data for ship contamination, then I request either evidence that ship contamination was not a problem during the entire study, or that the data be corrected and reanalyzed.

**General comments:**

The paper should be edited throughout for grammar. The authors use tense in confusing ways--they often use past tense when present tense is more appropriate and clear. For example, Lines 43-44: "The chemical components and sources of aerosol particles in the marine atmosphere *were* rather complicated" (italics mine). The chemical components and sources are still complicated, and the present tense should be used here. Please check tense use throughout. As well, verb endings are often incorrect. For example, line 25: "...*deriving* from the oxidation…". This should be "derived" here.

Define the size ranges meant by 'fine particles' and 'coarse particles' in this paper. Different studies use different definitions.

The authors must define each acronym when it is first used. For instance, MP1 and MP2 are brought up in lines 139-144 but are not explained or defined. Same for 'MA' (line 148).

There needs to be discussion in the methods of:
--Particle size range of the IGAC and SPAMS. In the results, there is mention that particles between .1-2.5 μm were considered from the SPAMS. Is this the size range used throughout the study? Not being able to capture particles <100 nm is a limitation of this study and should be acknowledged and discussed.

--Very important: how the authors corrected for potential contamination of pollution from the ship. Although pollution from the ship likely wouldn't impact the MSA measurements, it would impact the total aerosol population and mass concentrations. As well, contamination from ship pollution could alter the speciation of the particles, skewing the authors' results. Did the authors exclude time periods in which the ship pollution would have impacted the measurements?

--Length of the tubing used for sampling and whether there were corrections for particle and vapor losses within the tubing and associated uncertainties.

Line 125: 'and their populations' is confusing here. I see later in the text that particle populations are determined by the SPAMS. Please make that clear here.

Lines 145-147: The authors provide the MSA particle population. It would be of interest to provide the total particle population as well, for comparison. I see this comparison is made in section 3.3; perhaps the authors can refer the reader to that section for comparisons of MSA-containing particles to the total particle concentration.

Sections 3.3.1-3.3.7: For consistency and to provide a complete picture for each speciation, I suggest briefly including results from Figs 4 and 5 for each species subtype. Figs 4 and 5 are currently inconsistently discussed between sections. (For example, provide the mean fraction that each subtype contributes to the total population [Fig 4] within each section and so forth.) The authors are not limited by space for this journal, and this discussion is currently unsatisfying.

Section 3.3.7: discuss sources of NOx, HNO3 in the marine atmosphere here

**Figures/tables:**
Supplemental figures are not currently referenced in order in the main text. For instance, lines 68-72 jump from Figure S1 to Figure S4. Please update the SI figures to reflect the order they are referenced in the text.

Figure 3: it is very hard to read the speciation (Na+, $C_4H$, etc) on each mass spectra plot. I recommend increasing the font size, if possible.

Figure 4: It is very hard to read the legends on this plot. I highly recommend increasing the font size; there is likely enough space to make each legend into 2 columns. As well, the percents in the pie chart are difficult to read.

Figure S4: There need to be units on the colorbars for sea ice coverage (presumably percent) and Chl-a. As well, the units can be included in the figure caption (that should be updated to Fig. S4 instead of Fig. 4).

Figure 6. I quite like this schematic; all of the text could be larger for clarity.

**Technical comments:**
I suggest defining 'polynya' the first time it's mentioned in the abstract and main text, as it is not a common term.

Line 11: change to "lacking in knowledge"

Lines 17-18: do the authors mean that MSA uptake favored sea salt particles? Suggest rewording.

Line 104: W should be capitalized and there is a missing negative in the denominator of 'W/cm$^{-2}$'

Line 116: 'cloud effect' is confusing, I suggest rewording. I assume the authors are referring to the loss of data due to clouds?

Line 119: please define 'centration data'

Line 179: Do the authors mean "was consistent" instead of "consisted"?

Line 278: do the authors mean 'species' rather than 'particles'?

Line 284: the word 'except' here is confusing. It is unclear to me what the authors intend by this statement.

---

## Short Comment (SC1) · 20 Nov 2019

A few previous studies examined uptake of gaseous MSA onto aerosol particles, and I would like to bring them up for the author to consider: [1] Tang, M. J., and Zhu, T.: Heterogeneous reactions of gaseous methanesulfonic acid with NaCl and sea salt particles, Sci. China Ser. B-Chem., 52, 93-100, 2009. [2] Tang, M. J., Li, M. Q., and Zhu, T.: Heterogeneous reactions of gaseous methanesulfonic acid with calcium carbonate and kaolinite particles, Sci. China-Chem., 53, 2657-2662, 2010. There are a few more, and please refer to the two references for more information.

[Figure]

2019.

---

## Short Comment (SC2) · 21 Nov 2019

Thanks very much for providing the meaningful previous studies. I have viewed the references carefully. The experiments of heterogeneous condensation of gaseous MSA on NaCl, sea salt particles, calcium carbonate and kaolinite particles were very important to understand the uptake of MSA on different particles. It concluded that the uptake coefficient of gaseous MSA onto the NaCl and sea salt particles were too low, however, high MS- concentrations in the aerosols were observed in the marine atmosphere. It indicated the reactive uptake of DMS on existing particles would be the major route of the particulate MSA formation in the field marine atmosphere. That is the rea-

son caused the discrepancy of MSA uptake on different particles. Further simulation experiments on the reactive uptake of DMS on different particles are required to clarify the mechanism of the oxidation of DMS on different particles. The previous studies provided important information on the conversion of gaseous MSA to particulate MSA, which is useful to update the knowledge of the manuscript.
* * *

---

## Referee Comment (RC2) · Anonymous Referee #1 · 17 Dec 2019

Summary:

The manuscript "Uptake selectivity of Methanesulfonic Acid (MSA) on fine particles over polynya regions of the Ross Sea, Antarctica" presents results form a field campaign undertaken in 2017/2018. The focus of this study is to investigate the uptake of MSA on different particle types. For this purpose MSA mass concentrations and total aerosol population, coupled with size resolved data, were collected simultaneously. The results of this study provide a clear characterisation of MSA uptake in the presence of different pre-existing particles, which I believe is of high interest for the aerosol community. Therefore, I recommend the paper for publication after the following comments

have been addressed:

Major comment:

I think the results are described in a very clear and precise way. The only thing I was wondering about is the probability of the source of certain particles. It is discussed that Na and Mg are typically associated with sea spray aerosols, while EC and OC are more associated with primary emissions from combustion processes and K with biomass burning. Did the authors try to check where the air-masses were originating from during the campaigns to assess whether biomass-burning or in general combustion processes would have been expected during this period? If there were no known sources of such processes during this time – could this indicate that the pre-existing particles originated from long-range transport? I would recommend to include a discussion on this in the revised manuscript.

Minor comments:

1. The manuscript currently presents data described in a mixture of present and past tense. I recommend sticking to one tense throughout the manuscript.

2. The manuscript contains several mistakes regarding singular/plural expressions that should be revised.

3. It is stated in the manuscript that sea spray aerosols generated by bursting bubbles are generally in the course mode (page 15, lines 322-323). This is not correct as the majority of particles form bubble busting (considering number concentrations) peak at diameters around 100 nm. See for example De Leeuw et al. (2011) or Prather et al. (2013).

Specific comments:

Page 2, line 38: replace "have showed" with "have shown"

Page 2, line 43: replace "were" with "are"
Page 3, line 64: replace "intensity" with "intense"

Page 3, lines 71-72: replace "ices" with "ice" Line 72: As an example of the "minor comment 1": replace "have" with "had"

Page 5, line 104: use a capital "W" for the unit "Watt"

Page 6, line 139: remove the "the" in front of "leg I".

Page 6, line 414: replace "following" with "followed"

Page 7, line 158: replace "were presented" with "were present"; this mistake occurs more often in the manuscript.

Page 8, line 180: I am not sure I would call a R2=65 a "strong positive correlation", rather just a "positive correlation"

Page 10, line 232: Rephrase the beginning of the sentence – "The simultaneous..."

Page 11, line 234: Replace "suggesting" with "suggest"

Page 11, line 235: Add "c,d" to the citation of the figure 5

Page 11, line 251: rephrase "a few signals of .. "

Page 12, line 257: example of "minor comment 2": replace "were" with "was"

Page 14: line 299: replace "conforming" with "confirming"

Page 17: line 350: rephrase sentence starting with "The other one halogen radicals..."

Page 17: line 364: rephrase sentence starting with "Following by the MSA-Na..."

Page 17: lines 369-371: Delete last sentence as it is repeated in the conclusion.

References:

de Leeuw, G., Andreas, E. L, Anguelova, M. D., Fairall, C. W., Lewis, E. R., O'Dowd, C., Schulz, M., and Schwartz, S. E. (2011), Production flux of sea spray aerosol, Rev.
Geophys., 49, RG2001, doi:10.1029/2010RG000349.

Prather, K.A., T.H. Bertram, V.H. Grassian, G.B. Deane, M.D. Stokes, P.J. DeMott, L.I. Aluwihare, B.P. Palenik, F. Azam, J.H. Seinfeld, R.C. Moffet, M.J. Molina, C.D. Cappa, F.M. Geiger, G.C. Roberts, L.M. Russell, A.P. Ault, J. Baltrusaitis, D.B. Collins, C.E. Corrigan, L.A. Cuadra-Rodriguez, C.J. Ebben, S.D. Forestieri, T.L. Guasco, S.P. Hersey, M.J. Kim, W.F. Lambert, R.L. Modini, W. Mui, B.E. Pedler, M.J. Ruppel, O.S. Ryder, N.G. Schoepp, R.C. Sullivan, and D. Zhao (2013). Bringing the ocean into the laboratory to probe the chemical complexity of sea spray aerosol. PNAS, 110(19):7550–7555.

---

## Author Comment (AC1) · 30 Dec 2019

Summary/recommendations: This is an interesting paper that provides a generally clear and thorough analysis of particulate-phase MSA in the Ross Sea region. The authors have done an excellent job presenting several different results that provide a clear picture of MSA-containing particles in this region for particles between $\sim$0.1-2 $\mu$m. I particularly appreciated their discussions upon why MSA condenses on some particle types but not others. This study can become a useful resource to the community. However, there were details missing in the methods that made a complete evaluation of this study difficult. My primary concern is that the authors did not make it clear

whether the datasets were screened for ship exhaust (pollution) contamination. If this was not done, the results would likely be skewed, in particular those that discuss particle speciation. I have outlined a few other concerns in my review below. If the authors have corrected their data for ship contamination and simply didn't include these details in the manuscript, then I recommend that this study be published after the revisions discussed below. If the authors have not corrected their data for ship contamination, then I request either evidence that ship contamination was not a problem during the entire study, or that the data be corrected and reanalyzed. Thanks very much for the comments. The shipboard observations are a challenge and specifically in the marine environment. It is the case that ship emissions may impact the observation data. In this study, to minimize the impact of self-contaminations of the vessel on the observation results, the air inlet connecting to the monitoring instruments is fixed to a mast at 20 meters above the sea surface located at the bow of the R/V. Note that the major pollution source is from the chimney, which is located at the stern of the R/V and about 25 meters above the sea level. Hence, the pollution emissions from the vessel mainly located at the downwind of the sampling inlet, especially when the vessel is running. As high-time-resolution observations are used in this study, the self-contaminations from the vessel have been eliminated from the measurement results. The wind speed and wind directions were also monitoring during the observation period, which were used to determine if the observations were affected by the self-contaminations or not in this study. The data have been corrected to eliminate the impact of ship contamination in this study.

General comments: The paper should be edited throughout for grammar. The authors use tense in confusing ways–they often use past tense when present tense is more appropriate and clear. For example, Lines 43-44: "The chemical components and sources of aerosol particles in the marine atmosphere were rather complicated" (italics mine). The chemical components and sources are still complicated, and the present tense should be used here. Please check tense use throughout. As well, verb endings are often incorrect. For example, line 25: "... deriving from the oxidation...".

This should be "derived" here. Thanks for the suggestion, we have revised the tense throughout the manuscript.

Define the size ranges meant by 'fine particles' and 'coarse particles' in this paper. Different studies use different definitions. In this study the size range of 'fine particles' is from 0.1 to 2.0$\mu$m, size range of 'submicron particles' is from 0.1 to 1.0$\mu$m, and the size range of 'coarse particles' is from 1.0 to 2.0$\mu$m. We have added the definition in the manuscript.

The authors must define each acronym when it is first used. For instance, MP1 and MP2 are brought up in lines 139-144 but are not explained or defined. Same for 'MA' (line 148). MP1 and MP2 represent the high MSA population regions, MA represents the high MSA mass region. We have added the description in the manuscript.

There needs to be discussion in the methods of: –Particle size range of the IGAC and SPAMS. In the results, there is mention that particles between .1-2.5 $\mu$m were considered from the SPAMS. Is this the size range used throughout the study? Not being able to capture particles <100 nm is a limitation of this study and should be acknowledged and discussed. The measurement particle size range of $\sim$10$\mu$m for IGAC, and 0.1$\sim$2.5$\mu$m for SPAMS. It is true that particles smaller than 100nm cannot be detected by the SPAMS in this study. Note that most of the MSA particles were in the range of 0.1 to 1.0$\mu$m in the marine atmosphere (Ayers et al., 1997), indicating that the MSA particles measured in this study represent most of the MSA particles in the marine atmosphere. We have added the discussion in the manuscript.

–Very important: how the authors corrected for potential contamination of pollution from the ship. Although pollution from the ship likely wouldn't impact the MSA measurements, it would impact the total aerosol population and mass concentrations. As well, contamination from ship pollution could alter the speciation of the particles, skewing the authors' results. Did the authors exclude time periods in which the ship pollution would have impacted the measurements? As mentioned above, to minimize the impact

of self-contaminations of the vessel on the observation results in this study. The following methods were used: 1) The air inlet connecting to the monitoring instruments was fixed to a mast at 20 meters above the sea surface located at the bow of the R/V. Note that the major pollution source is from the chimney, which is located at the stern of the R/V and about 25 meters above the sea level. Hence, the pollution emissions from the vessel mainly located at the downwind of the sampling inlet, especially when the vessel was running. 2) The wind speed and wind directions were also monitoring during the observation period, which were used to determine if the observations were affected by the self-contaminations or not in this study. 3) As high-time-resolution observations were used in this study, the self-contaminations from the vessel were eliminated from the measurement results. In this study, the NOx concentration was also monitored simultaneously during the cruise (Fig. SS1). The NOx concentrations were extremely low and remained stable in this study, indicating that the sampling gases were rarely affected by the ship emissions. We have checked the data when high NOx concentrations were present. The data have been excluded, when the observations were impacted by the ship pollution in this study.

–Length of the tubing used for sampling and whether there were corrections for particle and vapor losses within the tubing and associated uncertainties. The length of the tubing is about 20 meter. It is true that the particle and vapor would lose in the tubing. In this study, conductive silicon tubing was used to minimize the particle and vapor lost in the tubing. A high velocity sampling system was also used in this study with a gas velocity of about 4.25m/s in the tubing. The residence time of the gases in the tubing is about 4.7 seconds. The measurements of aerosol particles with and without the tubing have been carried out. There are few differences between the measuring results with and without the tubing. Hence, the particle and vapor losses in the tubing can be neglected in this case.

Line 125: 'and their populations' is confusing here. I see later in the text that particle populations are determined by the SPAMS. Please make that clear here. We have

added the description in the manuscript.

Lines 145-147: The authors provide the MSA particle population. It would be of interest to provide the total particle population as well, for comparison. I see this comparison is made in section 3.3; perhaps the authors can refer the reader to that section for comparisons of MSA-containing particles to the total particle concentration. Thanks for the suggestion, we have added the notice to refer the reader to the section for comparisons of MSA-containing particles to the total particle concentration.

Sections 3.3.1-3.3.7: For consistency and to provide a complete picture for each speciation, I suggest briefly including results from Figs 4 and 5 for each species subtype. Figs 4 and 5 are currently inconsistently discussed between sections. (For example, provide the mean fraction that each subtype contributes to the total population [Fig 4] within each section and so forth.) The authors are not limited by space for this journal, and this discussion is currently unsatisfying. Thanks for the suggestion. In this study, we focused on the characteristics of MSA particles, hence, we did not discuss other species in this manuscript to make the manuscript clear and concise. Here, we provided the complete picture for each speciation in the SI (seen in Fig. S6). The mean fraction of each subtype contributes to the total population during leg I and leg II are also illustrated in the SI (seen in Fig. S8). We have added the discussion in this section to smooth the discussion between Fig 4 and Fig. 5.

Section 3.3.7: discuss sources of NOx, HNO3 in the marine atmosphere here We have added the discussion of NOx, HNO3 sources in the marine atmosphere in section 3.3.7.

Figures/tables:

Supplemental figures are not currently referenced in order in the main text. For instance, lines 68-72 jump from Figure S1 to Figure S4. Please update the SI figures to reflect the order they are referenced in the text. We have updated the Figure order in SI.

Figure 3: it is very hard to read the speciation (Na+, C 4 H, etc) on each mass spectra plot. I recommend increasing the font size, if possible. We have increased the font size in Figure 3.

Figure 4: It is very hard to read the legends on this plot. I highly recommend increasing the font size; there is likely enough space to make each legend into 2 columns. As well, the percents in the pie chart are difficult to read. We have enlarged the legend and the percents in the pie chart in Figure 4.

Figure S4: There need to be units on the colorbars for sea ice coverage (presumably percent) and Chl-a. As well, the units can be included in the figure caption (that should be updated to Fig. S4 instead of Fig. 4). We have revised in the Figure S4 in the SI.

Figure 6. I quite like this schematic; all of the text could be larger for clarity. Thanks very much, we have enlarged the text size in the Figure 6.

Technical comments:

I suggest defining 'polynya' the first time it's mentioned in the abstract and main text, as it is not a common term. 'polynya'is an area of open sea water surrounded by ice. We have added the definition of 'polynya' in the abstract and main text.

Line 11: change to "lacking in knowledge" We have changed in Line 11.

Lines 17-18: do the authors mean that MSA uptake favored sea salt particles? Suggest rewording. We have revised in the manuscript.

Line 104: W should be capitalized and there is a missing negative in the denominator of'W/cm -2 ' We have revised in the manuscript.

Line 116: 'cloud effect'is confusing, I suggest rewording. I assume the authors are referring to the loss of data due to clouds? It is the case that the satellite data of Chl-a is affected by the clouds, resulting in the loss of data. We have revised in the manuscript.

Line 119: please define 'centration data' We have checked the information. This is a typo here. It is ' We used the sea ice concentration data from the daily...'.

Line 179: Do the authors mean "was consistent" instead of "consisted"? We have changed in the manuscript.

Line 278: do the authors mean 'species' rather than 'particles'? For SPAMS detection, particle is determined individually to provide single particle chemical compositions and size. Hence, different types of particles can be identified (such as Na, Mg, SO42- etc.). Here, we obtained the particle count of different types of particle but did not the particle mass concentration with SPAMS. Hence, 'particles' was used here.

Line 284: the word 'except' here is confusing. It is unclear to me what the authors intend by this statement. We have revised in the manuscript.

References

Ayers, G.P., Cainey, J.M., Gillett, R.W., Ivey, J.P. Atmospheric sulphur and cloud condensation nuclei in marine air in the Southern Hemisphere, Phil. Trans. R. Soc. Lond. B, 1997, 352, 203-211

Please also note the supplement to this comment:
https://www.atmos-chem-phys-discuss.net/acp-2019-811/acp-2019-811-AC1-supplement.pdf
* * *
[Figure]

Fig. SS1 Time series of NO$_x$ concentration during the observation

**Fig. 1.**

**Supplement:**

**Supplementary Information**

**Uptake selectivity of Methanesulfonic Acid (MSA) on fine particles over polynya regions of Ross Sea, Antarctica**

Jinpei Yan[1,2*], Jinyoung Jung[3], Miming Zhang[1,2], Federico Bianchi[4], Yeejun Tham[4], Suqing Xu[1,2], Qi Lin[1,2], Shuhui Zhao[1,2], Lei Li[5], Liqi Chen[1,2]

*1 Key Laboratory of Global Change and Marine-Atmospheric Chemistry, Xiamen 361005, China;*

*2 Third Institute of Oceanography, Ministry of Natural Resources, Xiamen 361005, China;*

*3 Korea Polar Research Institute, 26 Songdomirae-ro, Yeonsu-gu, Incheon 21990, Republic of Korea;*

*4 Institute for Atmospheric and Earth System Research; University of Helsinki, 00014, Finland;*

*5 Institute of Mass Spectrometer and Atmospheric Environment, Jinan University, Guangzhou 510632, China*

Supplementary Material

Fig. S1. The cruise tracks of the observation in the Ross Sea (RS). Leg I was carried out during the early December (December 2 to 20, 2017). Leg II was carried out from middle January to February 14, 2018 in the RS, covering a large regions of 50°S to 78°S, 160°E to 185°E.

[Figure]

Fig. S2. Spatial distribution of sea ice and Chl-a concentrations. (a) Sea ice concentrations during December 2 to 20; (b) Sea ice concentrations during January 13 to February 14, 2018; (c) Mean Chl-a concentrations during Dec.4 to 14, 2017; (d) Mean Chl-a concentrations during Jan. 25 to Feb. 4, 2018.

[Figure]

Fig. S3. Gases and aerosols monitoring system. An underway aerosols monitoring system were deployed on the R/V "Xuelong" to carried out the observation in the Southern Ocean (SO). An In-situ Gas and Aerosol Composition monitoring system was used to determine the gaseous and aerosol water soluble ions, A Single Particle Aerosol Mass Spectrometer were used to determine the particle size distribution and chemical compositions.

[Figure]

Fig. S4. Calibration curves of MSA, chloride, sulfate and sodium for IGAC monitoring system. (a) Six out of eight concentrations of standard solutions (0.1-1000 ug/L) were selected for MSA calibration ($r^2$=0.998); (b) Six out of eight concentrations of standard solutions (0.1-2000 ug/L) were selected for Chloride calibration ($r^2$=0.997); (c) Six out of eight concentrations of standard solutions (0.1-4000 ug/L) were selected for Sulfate calibration ($r^2$=0.997); (d) Six out of eight concentrations of standard solutions (0.1-2000 ug/L) were selected for Sodium calibration ($r^2$=0.998).

[Figure]

Fig. S5. Relationship between MSA population and MSA mass concentration. (a) Correlation between MSA mass concentrations and MSA particle population; (b) Correlation between MSA particle population and total particle population.

[Figure]

Fig. S6 Average mass spectra of different particle types during the cruise. (a) Na, (b) Mg, (c) $SO_4^{2-}$,

(d) K, (e) OC, (f) EC, (g) $NO_x^-$ and (h) MSA.

[Figure]

Fig. S7. Correlation between MSA-Na and MSA-Mg particle population.

[Figure]

Fig. S8. Mean fraction of MSA sub-types to the total particle population during leg I and leg II.

[Figure]

---

## Author Comment (AC2) · 30 Dec 2019

Major comment: I think the results are described in a very clear and precise way. The only thing I was wondering about is the probability of the source of certain particles. It is discussed that Na and Mg are typically associated with sea spray aerosols, while EC and OC are more associated with primary emissions from combustion processes and K with biomass burning. Did the authors try to check where the air-masses were originating from during the campaigns to assess whether biomass-burning or in general combustion processes would have been expected during this period? If there were no known sources of such processes during this time–could this indicate that the preexisting particles originated from long-range transport? I would recommend to include a discussion on this in the revised manuscript. It is true that aerosol particles would be impacted by the long-rang transport sources. The back trajectories along the cruise tracks in Ross Sea are given in Fig. SS1. The major air masses originated from the local sources during the cruise. The aerosol particle chemical compositions did not reveal an obvious correlation with air back trajectories in this study (Fig. 4 and Fig. SS1). Hence, in this study, particles are mainly associated with the local sources but not the long-range transport. We have added the discussion in the manuscript. OC particles are often associated with anthropogenic sources, such as vehicle and coal combustion (Silva et al., 2000; Stiaras et al., 2008), marine biogenic sources (Quinn et al., 2014) and secondary sources (Horne et al., 2018). It is the case that OC particles are mainly derived from fossil fuel combustion and secondary sources in the coastal and urban regions. But in the marine atmosphere, OC particles are often determined by the marine biogenic sources (Quinn et al., 2014, Yan et al., 2018). EC particles are typically associated with primary emissions from fossil fuel combustion, such as ship emissions in the ocean area. K is often used as a marker of biomass burning in the continent, but K can also be derived from other sources, such as coal combustion, biological materials. In this study, the signature of K is very different from the K signature from biomass burning, indicating that K particles are not associated with the biomass burning. Na and Mg are often associated with sea salt particles in the marine atmosphere. Positive corrections between Na, Mg and wind speeds are present in the Fig. SS2, indicating that those particles are derived from sea spray aerosols. We have added the discussion in the manuscript.

Minor comments: 1. The manuscript currently presents data described in a mixture of present and past tense. I recommend sticking to one tense throughout the manuscript. Thanks for the suggestion, present tense is accepted. We have revised throughout the manuscript.

2. The manuscript contains several mistakes regarding singular/plural expressions that

should be revised. We have revised in the manuscript.

3. It is stated in the manuscript that sea spray aerosols generated by bursting bubbles are generally in the course mode (page 15, lines 322-323). This is not correct as the majority of particles form bubble busting (considering number concentrations) peak at diameters around 100 nm. See for example De Leeuw et al. (2011) or Prather et al. (2013). It is the case that sea spray aerosols peak at diameters around 100 nm in some studies (De Leeuw et al. (2011) or Prather et al. (2013)), and the major sea spray aerosols are in submicron size (number concentrations). But the sea salt particles have a wide size distribution, ranging from 0.01-8 $\mu$m (Clarke et al., 2006). The expression is ambiguous here. We have revised in the manuscript. Thanks for the suggestion.

Specific comments: Page 2, line 38: replace "have showed" with "have shown" We have revised in the manuscript.

Page 2, line 43: replace "were" with "are" We have revised in the manuscript.

Page 3, line 64: replace "intensity" with "intense" "intense" is accepted in the manuscript.

Page 3, lines 71-72: replace "ices" with "ice" Line 72: As an example of the "minor comment 1": replace "have" with "had" We have revised in the manuscript.

Page 5, line 104: use a capital "W" for the unit "Watt" We have revised in the manuscript.

Page 6, line 139: remove the "the" in front of "leg I". "the" has been removed in the manuscript.

Page 6, line 414: replace "following" with "followed" "followed" is accepted in the manuscript.

Page 7, line 158: replace "were presented" with "were present"; this mistake occurs more often in the manuscript. We have revised throughout the manuscript.

Page 8, line 180: I am not sure I would call a R2=65 a "strong positive correlation", rather just a "positive correlation" "positive correlation" is appropriate here.

Page 10, line 232: Rephrase the beginning of the sentence – "The simultaneous..." We have rephrased in the manuscript.

Page 11, line 234: Replace "suggesting" with "suggest" We have revised in the manuscript.

Page 11, line 235: Add "c,d" to the citation of the figure 5 We have added "c,d" to the citation of the Figure 5.

Page 11, line 251: rephrase "a few signals of.." We have revised in the manuscript.

Page 12, line 257: example of "minor comment 2": replace "were" with "was" We have revised in the manuscript.

Page 14: line 299: replace "conforming" with "confirming" We have revised in the manuscript. Page 17: line 350: rephrase sentence starting with "The other one halogen radicals..." We have revised in the manuscript.

Page 17: line 364: rephrase sentence starting with "Following by the MSA-Na..." We have rephrased in the manuscript.

Page 17: lines 369-371: Delete last sentence as it is repeated in the conclusion. We have deleted the sentence.

References: De Leeuw, G., Andreas, E. L, Anguelova, M. D., Fairall, C. W., Lewis, E. R., O'Dowd, C., Schulz, M., and Schwartz, S. E. (2011), Production flux of sea spray aerosol, Rev Geophys., 49, RG2001, doi:10.1029/2010RG000349. Prather, K.A., T.H. Bertram, V.H. Grassian, G.B. Deane, M.D. Stokes, P.J. DeMott, L.I. Aluwihare, B.P. Palenik, F. Azam, J.H. Seinfeld, R.C. Moffet, M.J. Molina, C.D. Cappa, F.M.Geiger,G.C.Roberts,L.M.Russell,A.P.Ault,J.Baltrusaitis,D.B.Collins,C.E.Corrigan, L.A. Cuadra-Rodriguez, C.J. Ebben, S.D. Forestieri, T.L. Guasco, S.P. Hersey, M.J.

Kim, W.F. Lambert, R.L. Modini, W. Mui, B.E. Pedler, M.J. Ruppel, O.S. Ryder, N.G. Schoepp, R.C. Sullivan, and D. Zhao (2013). Bringing the ocean into the laboratory to probe the chemical complexity of sea spray aerosol. PNAS, 110(19):7550– 7555. Clarke, A. D., Owens, S. R. Zhou, J. An ultrafine sea-salt flux from breaking waves: implications for cloud condensation nuclei in the remote marine atmosphere. J. Geophys. Res. 111, D06202. (doi:10.1029/2005JD006565), 2006. Silva, P. J., Carlin, R. A., Prather, K. A. Single particles analysis of suspended soil dust from Southern California. Atmos. Environ. 34, 1811-1820, 2000. Sitaras, I. E., Siskos, P. A. The role of primary and secondary air pollutants in atmospheric pollution: Athens urban area as a case study. Environ. Chem. Lett. 6, 59-69, 2008. Horne, J.R., Zhu, S., Montoya-Aguilera, J., Hinks, M.L., Wingen, L.M., Nizkorodov, S.A., Dabdub,D. Reactive uptake of ammonia by secondary organic aerosols: Implications for air quality. Atmos. Environ. 189, 1-8, 2018. Quinn, P.K., Bates, T.S., Schulz, K.S., Coffman, D.J., Frossard, A.A., Russell, L.M., Keene, W.C., Kieber, D.J. Contribution of sea surface carbon pool to organic matter enrichment in sea spry aerosol. Nature Geos. 7, 228-232, 2014. Yan, J., Lin, Q., Zhao, S., Chen, L., Li, L. Impact of marine and continental sources on aerosol characteristics using an on-board SPAMS over Southeast Sea, China. Environ. Sci. Pollution Res. 25, 30659-30670, 2018.

[Figure]

Fig. SS1 The back trajectories along the cruise tracks in Ross Sea

**Fig. 1.**

[Figure]

Fig. SS2 Time series of Na and Mg concentrations and wind speeds during the cruise.

**Fig. 2.**

---

## Author Response (AR1)

**Response to referee comments**

**Referee comments 1**

**Summary/recommendations:**

This is an interesting paper that provides a generally clear and thorough analysis of particulate-phase MSA in the Ross Sea region. The authors have done an excellent job presenting several different results that provide a clear picture of MSA-containing particles in this region for particles between ~0.1-2 μm. I particularly appreciated their discussions upon why MSA condenses on some particle types but not others. This study can become a useful resource to the community. However, there were details missing in the methods that made a complete evaluation of this study difficult. My primary concern is that the authors did not make it clear whether the datasets were screened for ship exhaust (pollution) contamination. If this was not done, the results would likely be skewed, in particular those that discuss particle speciation. I have outlined a few other concerns in my review below. If the authors have corrected their data for ship contamination and simply didn't include these details in the manuscript, then I recommend that this study be published after the revisions discussed below. If the authors have not corrected their data for ship contamination, then I request either evidence that ship contamination was not a problem during the entire study, or that the data be corrected and reanalyzed.

Thanks very much for the comments. The shipboard observation is a challenge and specifically in the marine environment. It is the case that ship emissions may impact the observation data. In this study, to minimize the impact of self-contaminations of the vessel on the observation results, the air inlet connecting to the monitoring instruments is fixed to a mast at 20 meters above the sea surface located at the bow of the R/V. Note that the major pollution source is from the chimney, which is located at the stern of the R/V and about 25 meters above the sea level. Hence, the pollution emissions from the vessel mainly located at the downwind of the sampling inlet, especially when the vessel is running. As high-time-resolution observations are used in this study, the self-contaminations from the vessel have been eliminated from the measurement results. The wind speeds and wind directions were also monitoring during the observation period, which were used to determine if the observations were affected by the self-contaminations or not in this study. The data has been corrected to eliminate the impact of ship contaminations in this study.

**General comments:**

The paper should be edited throughout for grammar. The authors use tense in confusing ways--they often use past tense when present tense is more appropriate and clear. For example, Lines 43-44: "The chemical components and sources of aerosol particles in the marine atmosphere were rather complicated" (italics mine). The chemical components and sources are still complicated, and the present tense should be used here. Please check tense use throughout. As well, verb endings are often incorrect. For example, line 25: "... deriving from the oxidation…". This should be "derived" here.

Thanks for the suggestion, we have revised the tense throughout the manuscript.

Define the size ranges meant by 'fine particles' and 'coarse particles' in this paper. Different studies use different definitions.

In this study the size range of 'fine particles' is from 0.1 to 2.0μm, seen in line 98 in the revised manuscript. Size range of 'submicron particles' is 0.1-1.0μm (line 243 in the revised manuscript), and the size range of 'coarse particles' is from 1.0 to 2.0μm (line 244 in the revised manuscript). We have added the definition in the manuscript.

The authors must define each acronym when it is first used. For instance, MP1 and MP2 are brought up in lines 139-144 but are not explained or defined. Same for 'MA' (line 148).

MP1 and MP2 represent the high MSA population regions (line 144-145), MA represents the high MSA mass region (line 152). We have added the description in the manuscript.

There needs to be discussion in the methods of:
--Particle size range of the IGAC and SPAMS. In the results, there is mention that particles between .1-2.5 μm were considered from the SPAMS. Is this the size range used throughout the study? Not being able to capture particles <100 nm is a limitation of this study and should be acknowledged and discussed.

The measurement particle size range of ~10μm for IGAC, and 0.1~2.5μm for SPAMS. It is true that particles smaller than 100nm cannot be detected by the SPAMS in this study. Note that most of the MSA particles were in the range of 0.1 to 1.0μm in the marine atmosphere (Ayers et al., 1997), indicating that the MSA particles measured in this study represent most of the MSA particles in the marine atmosphere. We have added the discussion in the manuscript (line 328-331).

--Very important: how the authors corrected for potential contamination of pollution from the ship. Although pollution from the ship likely wouldn't impact the MSA measurements, it would impact the total aerosol population and mass concentrations. As well, contamination from ship pollution could alter the speciation of the particles, skewing the authors' results. Did the authors exclude time periods in which the ship pollution would have impacted the measurements?

As mentioned above, to minimize the impact of self-contaminations of the vessel on the observation results in this study. The following methods were used: 1) The air inlet connecting to the monitoring instruments was fixed to a mast at 20 meters above the sea surface located at the bow of the R/V. Note that the major pollution source is from the chimney, which is located at the stern of the R/V and about 25 meters above the sea level. Hence, the pollution emissions from the vessel mainly located at the downwind of the sampling inlet, especially when the vessel was running. 2) The wind speeds and directions were also monitored during the observation period, which were used to determine if the observations were affected by the self-contaminations or not in this study. 3) As high-time-resolution observations were used in this study, the self-contaminations from the vessel were eliminated from the measurement results, seen in line 77-80.

In this study, the $NO_x$ concentration was also monitored simultaneously during the cruise (Fig. SS1). The $NO_x$ concentrations were extremely low and remained stable in this study, indicating that the sampling gases were rarely affected by the ship emissions. We have checked the data when high $NO_x$ concentrations were present. The data have been excluded, when the observations were impacted by the ship pollution in this study.

[Figure]

Fig. SS1 Time series of $NO_x$ concentration during the observation

--Length of the tubing used for sampling and whether there were corrections for particle and vapor losses within the tubing and associated uncertainties.

The length of the tubing is about 20 meter. It is true that the particle and vapor would lose in the tubing. In this study, conductive silicon tubing was used to minimize the particle lost in the tubing. A high velocity sampling system was also used in this study with a gas velocity of about 4.25m/s in the tubing. The residence time of the gases in the tubing is about 4.7 seconds. The measurements of aerosol particles with and without the tubing have been carried out. There are few differences between the measuring results with and without the tubing. Hence, the particle and vapor losses in the tubing can be neglected in this case.

Line 125: 'and their populations' is confusing here. I see later in the text that particle populations are determined by the SPAMS. Please make that clear here.

We have added the revised in the manuscript (line 127 in the revised manuscript).

Lines 145-147: The authors provide the MSA particle population. It would be of interest to provide the total particle population as well, for comparison. I see this comparison is made in section 3.3; perhaps the authors can refer the reader to that section for comparisons of MSA-containing particles to the total particle concentration.

Thanks for the suggestion, we have added the notice to refer the reader to the section for comparisons of MSA-containing particles to the total particle concentration, seen in line 151-152.

Sections 3.3.1-3.3.7: For consistency and to provide a complete picture for each speciation, I suggest briefly including results from Figs 4 and 5 for each species subtype. Figs 4 and 5 are currently inconsistently discussed between sections. (For example, provide the mean fraction that each subtype contributes to the total population [Fig 4] within each section and so forth.) The authors are not limited by space for this journal, and this discussion is currently unsatisfying.

Thanks for the suggestion. In this study, we focused on the characteristics of MSA particles, hence, we did not discuss other species in this manuscript to make the manuscript clear and concise. Here, we provided the complete picture for each speciation in the SI (seen in Fig. S6). The mean fraction of each subtype contributes to the total population during leg I and leg II are also illustrated in the SI (seen in Fig. S8). We have added the discussion in this section to smooth the discussion between Fig 4 and Fig. 5, seen in 320-331 in the revised manuscript.

Section 3.3.7: discuss sources of NOx, $HNO_3$ in the marine atmosphere here

We have added the discussion of $NO_x$, $HNO_3$ sources in the marine atmosphere in section 3.3.7, seen in line 280-285.

Figures/tables:

Supplemental figures are not currently referenced in order in the main text. For instance, lines 68-72 jump from Figure S1 to Figure S4. Please update the SI figures to reflect the order they are referenced in the text.

We have updated the Figure order in SI.

Figure 3: it is very hard to read the speciation (Na+, C 4 H, etc) on each mass spectra plot. I recommend increasing the font size, if possible.

We have increased the font size in Figure 3.

Figure 4: It is very hard to read the legends on this plot. I highly recommend increasing the font size; there is likely enough space to make each legend into 2 columns. As well, the percents in the pie chart are difficult to read.

We have enlarged the legend and the percents in the pie chart in Figure 4.

Figure S4: There need to be units on the colorbars for sea ice coverage (presumably percent) and Chl-a. As well, the units can be included in the figure caption (that should be updated to Fig. S4 instead of Fig. 4).

We have revised in the Figure S4 in the SI.

Figure 6. I quite like this schematic; all of the text could be larger for clarity.

Thanks very much, we have enlarged the text size in the Figure 6.

Technical comments:

I suggest defining 'polynya' the first time it's mentioned in the abstract and main text, as it is not a common term.

'polynya' is defined as an area of open sea water surrounded by ice. We have added the definition of 'polynya' in the abstract (line 12) and main text (line 58).

Line 11: change to "lacking in knowledge"
We have changed in line 11.

Lines 17-18: do the authors mean that MSA uptake favored sea salt particles? Suggest rewording.
We have revised in the manuscript (line 18-19).

Line 104: W should be capitalized and there is a missing negative in the denominator of 'W/cm -2 '
We have revised in the manuscript (line 106).

Line 116: 'cloud effect' is confusing, I suggest rewording. I assume the authors are referring to the loss of data due to clouds?
It is the case that the satellite data of Chl-a is affected by the clouds, resulting in the loss of data. We have revised in the manuscript (line 118).

Line 119: please define 'centration data'
We have checked the information. This is a typo here. It is ' We used the sea ice concentration data from the daily…'. (line 121)

Line 179: Do the authors mean "was consistent" instead of "consisted"?
We have changed in the manuscript (line 185).

Line 278: do the authors mean 'species' rather than 'particles'?
For SPAMS detection, particle is determined individually to provide single particle chemical compositions and size. Hence, different types of particles can be identified (such as Na, Mg, $SO_4^{2-}$ etc.). Here, we obtained the particle count of different types of particle but did not the particle mass concentration with SPAMS. Hence, 'particles' was used here.

Line 284: the word 'except' here is confusing. It is unclear to me what the authors intend by this statement.
Here we mean that the particle population was not the only impact factor for the uptake of MSA (line 298-299). We have revised in the manuscript.

References:

Ayers, G.P., Cainey, J.M., Gillett, R.W., Ivey, J.P. Atmospheric sulphur and cloud condensation nuclei in marine air in the Southern Hemisphere, Phil. Trans. R. Soc. Lond. B, 1997, 352, 203-211

**Referee comments 2**

Summary:

The manuscript "Uptake selectivity of Methanesulfonic Acid (MSA) on fine particles over polynya regions of the Ross Sea, Antarctica" presents results form a field campaign undertaken in 2017/2018. The focus of this study is to investigate the uptake of MSA on different particle types. For this purpose MSA mass concentrations and total aerosol population, coupled with size resolved data, were collected simultaneously. The results of this study provide a clear characterization of MSA uptake in the presence of different pre-existing particles, which I believe is of high interest for the aerosol community. Therefore, I recommend the paper for publication after the following comments have been addressed:

Major comment:

I think the results are described in a very clear and precise way. The only thing I was wondering about is the probability of the source of certain particles. It is discussed that Na and Mg are typically associated with sea spray aerosols, while EC and OC are more associated with primary emissions from combustion processes and K with biomass burning. Did the authors try to check where the air-masses were originating from during the campaigns to assess whether biomass-burning or in general combustion processes would have been expected during this period? If there were no known sources of such processes during this time–could this indicate that the pre-existing particles originated from long-range transport? I would recommend to include a discussion on this in the revised manuscript.

It is true that aerosol particles would be impacted by the long-rang transport sources. The back trajectories along the cruise tracks in Ross Sea are given in the Fig. SS2. The major air masses originated from the local sources during the cruise. The aerosol particle chemical compositions did not reveal an obvious correlation with air back trajectories in this study (Fig. 4 and Fig. SS2). Hence, in this study, particles were mainly associated with the local sources but not the long-range transport.

OC particles are often associated with anthropogenic sources, such as vehicle and coal combustion (Silva et al., 2000; Stiaras et al., 2008), marine biogenic sources (Quinn et al., 2014) and secondary sources (Horne et al., 2018). It is the case that OC particles are mainly derived from fossil fuel combustion and secondary sources in the coastal and urban regions. But in the marine atmosphere, OC particles are often determined by the marine biogenic sources (Quinn et al., 2014, Yan et al., 2018). EC particles are typically associated with primary emissions from fossil fuel combustion, such as ship emissions in the ocean area. K is often used as a marker of biomass burning in the continent, but K can also be derived from other sources, such as coal combustion, biological materials. In this study, the signature of K is very different from the K signature from biomass burning, indicating that K particles are not associated with the biomass burning. Na and Mg are often associated with sea salt particles in the marine atmosphere. Positive corrections between Na, Mg and wind speeds are present in the Fig. SS3, indicating that those particles are derived from sea spray aerosols.

We have added the discussion in the manuscript, seen in section 3.3.1-3.3.7.

[Figure]

Fig. SS2 The back trajectories along the cruise tracks in Ross Sea

[Figure]

Fig. SS3 Time series of Na and Mg concentrations and wind speeds during the cruise.

Minor comments:

1. The manuscript currently presents data described in a mixture of present and past tense. I recommend sticking to one tense throughout the manuscript.

Thanks for the suggestion, present tense is accepted. We have revised throughout the manuscript.

2. The manuscript contains several mistakes regarding singular/plural expressions that should be revised.

We have revised in the manuscript.

3. It is stated in the manuscript that sea spray aerosols generated by bursting bubbles are generally in the course mode (page 15, lines 322-323). This is not correct as the majority of particles form bubble busting (considering number concentrations) peak at diameters around 100 nm. See for example De Leeuw et al. (2011) or Prather et al. (2013).

It is the case that sea spray aerosols peak at diameters around 100 nm in some studies (De Leeuw et al. (2011) or Prather et al. (2013)), and the major sea spray aerosols are in submicron size (number concentrations). But the sea salt particles have a wide size distribution, ranging from 0.01-8 μm (Clarke et al., 2006). The expression is ambiguous here. We have revised in the manuscript (line 348-349). Thanks for the suggestion.

Specific comments:

Page 2, line 38: replace "have showed" with "have shown"

We have revised in the manuscript, seen in line 38.

Page 2, line 43: replace "were" with "are"

We have revised in the manuscript, seen in line 43.

Page 3, line 64: replace "intensity" with "intense"

"intense" is accepted in the manuscript, seen in line 63.

Page 3, lines 71-72: replace "ices" with "ice" Line 72: As an example of the "minor comment 1": replace "have" with "had"

We have revised in the manuscript.

Page 5, line 104: use a capital "W" for the unit "Watt"

We have revised in the manuscript, seen in line 106.

Page 6, line 139: remove the "the" in front of "leg I".

The "the" has been removed in the manuscript.

Page 6, line 414: replace "following" with "followed"

The "followed" is accepted in the manuscript.

Page 7, line 158: replace "were presented" with "were present"; this mistake occurs more often in the manuscript.
We have revised throughout the manuscript.

Page 8, line 180: I am not sure I would call a $R^2$=65 a "strong positive correlation", rather just a "positive correlation"
"positive correlation" is appropriate here, line 186.

Page 10, line 232: Rephrase the beginning of the sentence – "The simultaneous..."
We have rephrased in the manuscript, seen in 239-241.

Page 11, line 234: Replace "suggesting" with "suggest"
We have revised in the manuscript, seen in line 241.

Page 11, line 235: Add "c,d" to the citation of the figure 5
We have added "c,d" to the citation of the Figure 5, seen in line 242.

Page 11, line 251: rephrase "a few signals of.."
We have revised in the manuscript, seen in line 260.

Page 12, line 257: example of "minor comment 2": replace "were" with "was"
We have revised in the manuscript, seen in line 266.

Page 14: line 299: replace "conforming" with "confirming"
We have revised in the manuscript, seen in line 314.
Page 17: line 350: rephrase sentence starting with "The other one halogen radicals..."
We have revised in the manuscript, seen in line 375-377.

Page 17: line 364: rephrase sentence starting with "Following by the MSA-Na..."
We have rephrased in the manuscript, seen in line 388.

Page 17: lines 369-371: Delete last sentence as it is repeated in the conclusion.
We have deleted the sentence in the manuscript.

We tried our best to improve the manuscript and revised carefully to improve the manuscript. Here we did not list the minor changes but marked in red in the revised paper. We appreciate for Editors/Reviewers' warm work earnestly, and hope that the correction will meet with approval. Once again, thank you very much for your comments and suggestions.

[revised manuscript text omitted]

---

## Author Response (AR2)

**Response to editor's comments**

The manuscript can be published after the grammar is thoroughly checked through out the manuscript.

There seems to be inconsistency in tense used in the manuscript. For example in page 6 line 127 it is written: "MSA mass concentration were measured..." and line 142 "...particle populations are determined...". You should use past tense when describing what was done and how. Also, as a general advice: when discussing your new results and observations presented in this manuscript you should use past tense (page 12 line 264: "... In this study, MSA-EC particles WERE characterized by strong peaks of...") instead of current tens. When you talk about the generally accepted and already published knowledge, you should use the current tense (page 12, line 266) "EC particles are often associated with ship emissions in the ocean atmosphere (Yan et al., 2018). " The tense should be corrected throughout the manuscript.

Thanks very much for the suggestion. We have checked the grammar through out the manuscript. Past tense is used to describe the results and observations, and current tense is used for the generally accepted and already published knowledge in the manuscript.